# Role of Adipose Tissue Hormones in Pathogenesis of Cryptoglandular Anal Fistula

**DOI:** 10.3390/ijms25031501

**Published:** 2024-01-25

**Authors:** Marcin Włodarczyk, Jakub Włodarczyk, Kasper Maryńczak, Anna Waśniewska-Włodarczyk, Urszula Doboszewska, Piotr Wlaź, Łukasz Dziki, Jakub Fichna

**Affiliations:** 1Department of General and Oncological Surgery, Faculty of Medicine, Medical University of Lodz, Pomorska 251, 92-213 Lodz, Poland; 2Department of Biochemistry, Faculty of Medicine, Medical University of Lodz, Mazowiecka 6/8, 92-215 Lodz, Poland; 3Department of Normal and Clinical Anatomy, Medical University of Lodz, Żeligowskiego 7/9, 90-752 Lodz, Poland; 4Department of Pharmacobiology, Jagiellonian University Medical College, Medyczna 9, 30-688 Kraków, Poland; 5Department of Animal Physiology and Pharmacology, Institute of Biological Sciences, Maria Curie-Skłodowska University, Akademicka 19, 20-033 Lublin, Poland

**Keywords:** cryptoglandular perianal fistula, adipose tissue hormones, cytokines, adipokines, proinflammatory factors

## Abstract

The cryptoglandular perianal fistula is a common benign anorectal disorder that is managed mainly with surgery and in some cases may be an extremely challenging condition. Perianal fistulas are often characterized by significantly decreased patient quality of life. Lack of fully recognized pathogenesis of this disease makes it difficult to treat it properly. Recently, adipose tissue hormones have been proposed to play a role in the genesis of cryptoglandular anal fistulas. The expression of adipose tissue hormones and epithelial-to-mesenchymal transition (EMT) factors were characterized based on 30 samples from simple fistulas and 30 samples from complex cryptoglandular perianal fistulas harvested during surgery. Tissue levels of leptin, resistin, MMP2, and MMP9 were significantly elevated in patients who underwent operations due to complex cryptoglandular perianal fistulas compared to patients with simple fistulas. Adiponectin and E-cadherin were significantly lowered in samples from complex perianal fistulas in comparison to simple fistulas. A negative correlation between leptin and E-cadherin levels was observed. Resistin and MMP2 levels, as well as adiponectin and E-cadherin levels, were positively correlated. Complex perianal cryptoglandular fistulas have a reduced level of the anti-inflammatory adipokine adiponectin and have an increase in the levels of proinflammatory resistin and leptin. Abnormal secretion of these adipokines may affect the integrity of the EMT in the fistula tract. E-cadherin, MMP2, and MMP9 expression levels were shifted in patients with more advanced and complex perianal fistulas. Our results supporting the idea of using mesenchymal stem cells in the treatment of cryptoglandular perianal fistulas seem reasonable, but further studies are warranted.

## 1. Introduction

Cryptoglandular perianal fistulas are considered a later, chronic stage of perianal abscesses. The glandular crypts at the level of the dentate line are the typical origin of fistulas, which are induced by purulent and inflammatory processes [1]. The cryptoglandular perianal fistula is a common benign anorectal disorder that is treated mainly with surgery. Unfortunately, despite proper therapy and management, this disease is characterized by a high recurrence rate. The surgical management of cryptoglandular perianal fistulas may be challenging, and the ideal approach remains questionable even among experienced colorectal surgeons. Currently, there is a lack of available surgical techniques that can be considered as the gold standard treatment of the perianal fistula [2]. Fistulotomy, as one of the most often used procedures, is associated with a healing rate of >90% [3]. However, patients treated with fistulotomy have a risk of developing postoperative perianal sphincter dysfunction, especially women or patients with complex fistulas, preoperative incontinence, recurrent disease, or previous anorectal surgeries [4]. A better understanding of the pathogenesis underlying the formation of chronic cryptoglandular perianal fistulas may improve the final outcomes; however, currently, the complex etiology of the disease still remains fully unknown, and an inflammatory process appears to play a crucial role [5].

The role of visceral adipose tissue (VAT) in colonic inflammation has been reported, and its abnormal accumulation is often associated with a more severe disease phenotype as well as a poor prognosis in perianal patients [6,7,8,9]. The VAT plays a key role in the synthesis of adipokines, serving as signaling molecules or cytokines. These bioactive substances play a vital role in the regulation of diverse physiological processes, encompassing metabolism, inflammation, and immune function. Among the adipokines frequently studied are adiponectin, resistin, leptin, and SRFP5. The literature indicates that the dysregulation of these molecules has been associated with a range of metabolic and inflammatory disorders [10].

Epithelial-to-mesenchymal transition (EMT) and the abnormal integrity of the extracellular matrix (ECM) have been described in several human physiological and pathological processes, and their potential role has been suggested in the development of cryptoglandular anal fistula formation [11,12]. The EMT and ECM are characterized by changes in cell morphology and behavior [13]. So far, there is an abundant number of molecular markers that reflect the shift in cellular characteristics. The most common and broadly investigated involve E-cadherin, β-catenin, tissue zinc-finger protein SNAI1 (Snail), MMP2, and MMP9.

The aim of this study was to characterize the role of adipose tissue hormones in the pathogenesis of cryptoglandular anal fistula, with the specific aims of characterizing the potential role of the proinflammatory, EMT, and ECM factors responsible for the development of chronic inflammation.

## 2. Results

The analyzed group included 31 men (51.7%) at a mean age of 41.4 ± 11.3 years and 29 women (48.3%) at a mean age of 44.8 ± 13.9 years. Table 1 presents detailed characteristics of the demographic data and the basic parameters assessed in the study in qualified patients with analysis of differences between the gender groups.

### 2.1. Tissue Adipokines

In our study, we analyzed the levels of adipokines in tissues collected from cryptoglandular fistulas. Patients with complex fistulas were characterized by significantly higher levels of leptin and resistin compared to patients with simple fistulas (respectively, 22 ± 8.07 ng/mL vs. 12.77 ± 9.3 ng/mL, *p* < 0.001; 2447 ± 1107 pg/mL vs. 1538 ± 1069 pg/mL, *p* = 0.002). No statistically significant differences were observed in SFRP5 level and complexity of perianal fistulas (simple fistula: 353.6 ± 263 pg/mL vs. complex fistula: 283.7 ± 173.4 pg/mL, *p* = 0.23). The tissue concentration of adiponectin was significantly higher in patients with simple fistulas compared to patients with complex fistulas (respectively, 9429 ± 5021 pg/mL vs. 4417 ± 2737 pg/mL, *p* < 0.001) (Figure 1).

### 2.2. Integrity of ECM and EMT

In our study, there were no statistically significant differences between simple and complex perianal fistula patients in levels of tissue zinc-finger protein SNAI1 (Snail) (515.1 ± 311.2 pg/mL vs. 542.9 ± 303.2 pg/mL, *p* = 0.73) and β-catenin (115.8 ± 70.13 ng/mL vs. 112.2 ± 69.19 ng/mL, *p* = 0.84). In the group of simple fistula patients, significantly lower levels of MMP2 and MMP9 were observed compared to complex fistula patients (respectively, 311.1 ± 218.5 ng/mL vs. 426.4 ± 200.1 ng/mL, *p* = 0.04; 4116 ± 2598 pg/mL vs. 5857 ± 2798 pg/mL, *p* = 0.02). Simple fistulas were characterized by significantly higher levels of E-cadherin when compared to complex fistulas (5217 ± 2969 pg/mL vs. 3469 ± 2175 pg/mL, *p* = 0.01) (Figure 2).

Individual variables were next analyzed for correlations with tissue concentrations of selected adipokines (leptin, resistin, and adiponectin) with E-cadherin, MMP2, and MMP9. The leptin levels negatively correlated with E-cadherin levels (r = −0.3493; *p* = 0.006). MMP2 and MMP9 correlated positively with tissue leptin level; however, statistical significance was not observed (respectively, r = 0.1568, *p* = 0.231 and r = 0.1237, *p* = 0.346). Similar trends were observed with resistin analysis. Resistin levels negatively correlated with E-cadherin and positively correlated with MMP2 and MMP9 levels (respectively, r = −0.0714, *p* = 0.588; r = 0.297, *p* = 0.021; and r = 0.0127, *p* = 0.924). Contrary observations were made regarding the adiponectin levels. Adiponectin positively correlated with E-cadherin levels and negatively correlated with MMP2 and MM9 levels (respectively, r = 0.2696, *p* = 0.037; r = −0.1182, *p* = 0.369; and r = −0.1072, *p* = 0.415) (Figure 3).

## 3. Discussion

In recent studies conducted in patients with a persistent anal fistulas after surgical intervention, an incomplete healing of the fistula was observed at the site of the original internal fistula opening [14]. This observation and the fact that there are, most likely, no factors predisposing to failure suggest that the persistence of anal fistulas is caused by ongoing inflammation in the remaining fistula tract. A recent study revealed that most cryptoglandular anal fistulas are lined with granulation tissue (75%), which is suggestive of an inflammatory response [11].

The epithelial-to-mesenchymal transition (EMT) has been described in normal and neoplastic processes, including wound healing. Recently, an abnormal integrity of the extracellular matrix (ECM) and EMT has been observed in several human physiological and pathological processes, and its potential role has been shown in the development of cryptoglandular anal fistulas and fistula formation related to Crohn’s disease [1,15,16].

The EMT, through stimulating chronic inflammation and the release of proinflammatory cytokines, could then lead to an increased migration of epithelial cells which could contribute to fistula formation [17]. Abnormal integrity of the ECM significantly impairs the healing of the fistula tract [1]. The formation of fistula tracks could also be facilitated by the release of other substances, such as matrix metalloproteinases (MMPs), which are enzymes that degrade the extracellular matrix and facilitate cell migration and whose secretion by macrophages and fibroblasts is increased in the EMT [18]. Once formed, the fistula could allow the entry of other bacteria, which could continue to feed this process.

The gelatinases A (MMP-2) and B (MMP-9) are found among others in the gastrointestinal tract and have been shown to be highly upregulated in the large intestine during inflammatory processes [18,19,20].

Decreased MMP-2 expression was observed in inflammatory bowel disease (IBD) patients and was hypothesized to deregulate the intestinal barrier and stimulate fibrosis [21]. Another study reported that MMP-2-deficient mice were more resilient to acute dextran sulfate sodium (DSS) colitis and were characterized by better clinical, micro-, and macroscopic outcomes [22]. On the other hand, MMP-2 and its increased expression were observed in perianal fistulas in Crohn’s disease. An excess of MMP-2 was linked to fistula creation and genesis [23]. In accordance with the discussed papers, we noted that patients with complex perianal fistulas were characterized by higher expression of MMP-2.

Multiple studies have indicated that the expression of MMP-9 is increased during episodes of acute inflammation in individuals with IBD [24,25]. In DSS-induced colonic injury animal models, deletion of MMP-9 has proven to be advantageous in attenuating colonic damage and inflammation induced by both S. typhimurium and DSS [26,27]. These findings support the theory that the presence of MMP-9 in the colon leads to changes in the composition of the fecal microbiome and significantly influences the development of bacteria-induced colitis in mice. Moreover, MMP-9 plays a role similar to MMP-2 in the formation of fistulas, but it may also contribute to tissue damage by activating neutrophils in areas of acute inflammation and by accelerating the breakdown of matrix proteins that have been partially degraded by other MMPs [21]. MMP-9 actively participates in the inflammatory response, impedes the epithelial repair process, delays wound healing, heightens endothelial permeability, and triggers the release of cytokines and chemokines, such as interleukin IL-1β, IL-8, and TGF-β [23,28]. In our study, higher expression of MMP-9 was observed in patients treated for more advanced and complex perianal fistulas.

Snail, a typical marker (transcriptional factors) in the EMT process, was found to be significantly overexpressed in CD patients compared to those in the control group [29]. Additionally, it was found that the expression of Snail was elevated in patients who had fistulas compared to those with healthy anal mucosa samples [30]. E-cadherin, which is a typical marker of epithelial cells, was exclusively present in healthy anal mucosa but was absent in the proximal part of the fistula and only minimally detectable in the distal part [30]. Furthermore, its expression was reduced in the inflamed mucosa of individuals with IBD when compared to healthy mucosa [29,31]. The decrease in E-cadherin expression, along with a robust presence of protein kinase RNA-like endoplasmic reticulum kinase (pERK) and NF-κB, two intracellular factors induced by TGF-β1, was also associated with the EMT process [31,32]. The strong expression of Snail and the absence of E-cadherin may serve as markers for identifying the occurrence of EMT [16]. Correspondingly, we observed a more enhanced EMT in patients with complex cryptoglandular perianal fistulas compared to patients with simple fistulas. A slight trend in the increase of Snail expression correlated with the complexity of the fistula, and E-cadherin was significantly lowered in the complex fistula group.

The Wnt signaling pathway mainly involves β-catenin, a central component of cadherin proteins that regulates cell adhesion and gene transcription. When Wnt ligands are absent, cytoplasmic β-catenin undergoes phosphorylation by glycogen synthase kinase-3β, leading to the subsequent proteasomal degradation of β-catenin [33]. Wnt signaling is typically active in response to normal developmental processes and tissue growth. In the context of colonic inflammation, tumor necrosis factor-alpha (TNFα) triggers the upregulation of Wnt/β-catenin target genes. Notably, TNFα-induced activation of the Wnt/β-catenin pathway serves a protective role against apoptosis [34]. These discoveries underline the critical involvement of TNFα-induced Wnt/β-catenin signaling in the process of wound healing within IBD. However, in our study we have not observed any differences between the expression of β-catenin and the complexity of perianal fistulas.

The lack of changes in the tissue levels of β-catenin while the cadherin concentration is decreased in complex perianal cryptoglandular fistulas could be attributed to the intricate regulatory mechanisms involved in the EMT, including TGF-β1-induced β-catenin stabilization, alternative pathways, and different stages of the EMT [35,36]. Further research and experimentation are needed to pinpoint the exact reasons for this specific observation.

Recent studies suggest that white adipose tissue (WAT), in addition to its ability to respond to afferent signals from the endocrine system and the central nervous system, also expresses and secretes factors with important functions [37]. In fact, WAT produces and releases a great number of multifunctional proteins. Those released exclusively by WAT are collectively named “adipokines”. Among these leptin, resistin, adiponectin, and SRFP5 appear to present a promising key role in the development and suppression of the inflammatory response, and their role has been confirmed in other inflammatory disorders, such as Crohn’s disease [1].

Leptin has a dual action in human organisms; it acts as a cytokine and as a hormone. As a hormone, it has an impact on bone metabolism and various endocrine functions. Moreover, leptin is crucial in thermoregulation. As a cytokine, leptin has proinflammatory activity [38]. In obese patients, an increased level of leptin was associated with the low-grade inflammatory state, which resulted in cardiovascular diseases, degenerative disease, type II diabetes, and autoimmune disease in this group of patients [39,40]. Sitaraman et al. documented that the ongoing inflammation in colonic epithelial cells stimulates the expression and release of leptin into the intestinal lumen [41]. Subsequently, luminal leptin serves as an activator of nuclear factor kappa B (NF-κB), a powerful stimulator of proinflammatory processes. This activation ultimately leads to damage in the epithelial wall and the infiltration of neutrophils. Furthermore, the migration of macrophages towards deceased cells is intensified, which subsequently amplifies the release of proinflammatory cytokines, including but not limited to IL-6, IL-1, and IL-12 [42]. In our study, leptin was more abundantly expressed in patients with complex perianal fistulas when compared to patients with simple fistulas.

Resistin was firstly described in 2001. Previously, increased serum resistin levels were reported in rodent models of obesity, diabetes, and lung inflammation [43,44]. In humans, macrophages, peripheral blood mononuclear cells, and bone marrow cells are the main source of circulating resistin [45]. Lower amounts of resistin were also described in cells from the gastrointestinal tract, mainly in the colonic epithelium [46]. Resistin activates inflammatory cytokines and stimulates the expression of cell adhesion molecules [47]. According to Tarkowski et al., the proinflammatory activity of resistin might be mediated through its connection to the endotoxin receptor toll-like receptor 4 [48]. Moreover, increased resistin levels have been observed in IBD patients and correlated with the disease activity score [49]. Correspondingly, we observed an increase in the expression of resistin in patients with complex cryptoglandular perianal fistulas compared to patients with simple fistulas.

Another plasma protein that is secreted by adipose tissue is adiponectin. Adiponectin plays a significant role in anti-inflammatory, antidiabetic, and antiatherogenic processes [50]. Unlike the above adipokines, adiponectin is a protective factor for cardiovascular diseases, type 2 diabetes, and obesity [51]. There are three types of ADPN receptors, T-cadherin, AdipoR1, and AdipoR2, which are located in the pancreas, muscles, liver, adipose tissue, heart, and brain [52]. Adiponectin achieves its anti-inflammatory activity by suppressing cytokines and the production of TNF α [53]. Adiponectin counteracts oxidative-stress-mediated cytotoxicity. In accordance, in our study the protective effect of adiponectin was observed. Patients with higher expression of adiponectin were characterized by simple perianal cryptoglandular fistulas.

The Wnt signaling pathway coordinates tissue homeostasis and development. The dysregulation of the Wnt pathway results in cancer, developmental defects, and degenerative disorders [54]. Secreted frizzled-related protein 5 (SFRP-5) belongs to the SFRP family, which shows a negative effect on the Wnt pathway. SFRP-5 inhibits inflammatory processes through the non-canonical Wnt5a/c-Jun N-terminal kinase (JNK) signaling pathway. This adipokine was reported to decrease cardiac inflammation and protect heart tissue against ischemia [55]. Moreover, SFRP-5 prevents the accumulation of activated macrophages and inhibits the activation of pro-inflammatory cytokines such as IL-6 and TNF-α through the JNK [56]. In our study, we observed a slight trend in the increased expression of SFRP-5 in samples gathered from patients with simple perianal fistula.

To the best of our knowledge, this is the first study to analyze the adipose tissue resected from a cryptoglandular perianal fistula. We present the possible mechanism of the disturbed healing of fistulas, which is based on the elevation in proinflammatory adipokines and the decrease in anti-inflammatory adiponectin, together with the alternation in concentrations of MMP2, MMP9, and E-cadherin.

## 4. Materials and Methods

### 4.1. Patients

This prospective clinical study was performed with adult patients with the diagnosis of cryptoglandular anal fistula hospitalized at the Department of General and Colorectal Surgery at the Medical University of Lodz, Poland. Within the 36 months of the study, we enrolled 60 adult patients of Caucasian origin with cryptoglandular anal fistulas, which were evaluated under anesthesia and by endoanal ultrasound (EAUS). All approvals required to perform this study were obtained from the Committee of Bioethics of the Medical University of Lodz (RNN/36/18/KE). The study enrolled only patients hospitalized with the diagnosis of cryptoglandular anal fistula who gave their written and informed consent to participate. Case report forms were collected for each qualified patient in the study and included demographic data, comorbidities, and clinical classification of perianal fistula.

### 4.2. Exclusion Criteria

Patients with extrasphincteric and subanodermal fistulas or anal fistulas related to Crohn’s disease were excluded. Other exclusion criteria included current smokers; patients with a history of cardiovascular, pulmonary, or kidney disease; patients with allergies, diabetes, inflammatory bowel diseases, lichen planus, psoriasis, atopic dermatitis, and other autoimmune skin lesions; those treated with anti-inflammatory drugs (except azathioprine and steroids), antioxidants, or statins (which can affect the inflammation process); and those treated with anti-depressants, sedatives, and hypnotics. Steroids and immunosuppressive drugs are commonly employed in the treatment of anal fistulas arising from Crohn’s disease [57]. It is crucial to recognize that the immunosuppressive nature of these medications can markedly impede the post-surgical healing process for cryptoglandular fistulas, potentially contributing to their persistence or recurrence [58]. Recent studies propose that an adjunctive approach targeting cytokines, specifically anti-inflammatory IL-1β, may offer benefits in treating anal fistulas. However, the clinical significance of these findings must undergo further investigation before practical application [59].

### 4.3. Complexity of Perianal Fistulas

All perianal fistulas and their complexity were classified according to the Standard Practice Task Force (SPTF) classification. Simple perianal fistulas included fistulas involving less than one-third of the sphincter and in which fistulotomy was possible without risk of incontinence. Complex perianal fistulas were characterized by a high risk of incontinence during fistulotomy and involved supralevator fistulas, multiple and horseshoe tract fistulas, and anterior fistulas in females. The SPTF classification was developed by the American Society of Colon and Rectal Surgeons [60].

### 4.4. Collection of Tissue Samples

To quantify and assess the analyzed molecules, forceps tissue samples from resected perianal fistulas were collected immediately after surgical operation. After isolation of tissue, the biopsy specimens were immediately frozen and kept at −80 °C until processing.

### 4.5. Assessment of Tissue Adipokines and the Integrity of the ECM and EMT

The tissue concentrations of leptin, adiponectin, resistin, SRFP5, Snail-1, B-catenin, E-cadherin, MMP2, and MMP9 were determined with quantitative sandwich enzyme- linked immunosorbent assays (ELISAs) using commercially available primary and enzyme-linked secondary antibodies (Elabscience, Houston, TX, USA). Tissue samples were minced and homogenized according to the manufacturer’s instructions. ELISA assays were performed with polystyrene, 96-well, flat-bottomed microtiter plates (Polgen, Lodz, Poland). Each well was coated with an antigen diluted in 100 mL of carbonate buffer, pH 9.6. Absorbance was read in a microplate reader at proper wavelength as listed in the kit’s instructions (MicroPlate Reader; Bio-Rad, Hercules, CA, USA). For each detection, calibration blank tests have been taken into account. Each determination was carried out in triplicate in accordance with the principles of the laboratory. Intra-assay coefficient of variability (CV) was analyzed by measuring the variance between data points within an assay on the same plate, meaning sample replicates ran within the same plate. In this study the ELISA test met the criteria of our laboratory, and inter-assay CVs were lower than 10%. All steps in the ELISA test were conducted at room temperature on an orbital shaker. During incubation, plates were placed in humid chambers to prevent sample evaporation. The optimal concentration for each examined protein and antibody was established by titration. The amount of antigen coating the plate and the optimal dilutions for the primary antibodies used were determined experimentally by performing calibration curves.

### 4.6. Statistical Analysis

The data were analyzed using GraphPad Prism 9 software. A Shapiro–Wilk test was used to test the determined normality of the distribution of variables. Continuous variables were expressed as median, minimum and maximum, or mean ± standard deviation; categorical data were described with absolute frequencies and percentages. Basic comparisons between groups were performed using the Student’s *t* test (or nonparametric Mann–Whitney U test) and Fisher’s exact test (or χ^2^ test), depending on the distribution of the variables. Correlations were evaluated using the Pearson’s test or Spearman’s rank correlation coefficient (r) test depending on the normality of the distribution. A value of *p* < 0.05 was considered statistically significant.

### 4.7. Study Limitations

The study has several limitations that need to be acknowledged. Firstly, the relatively small sample size precluded any subgroup analysis of the clinical outcomes. Additionally, the absence of an appropriate control tissue, as previously noted by others [28], is a notable constraint. It is worth noting that obtaining tissue from the intersphincteric space in a healthy patient is not feasible, and the use of cadaveric tissue is also impractical due to potential alterations in tissue content. This study constitutes a pioneering exploration in the field, leveraging preliminary results and employing a singular technique, specifically ELISA of digested tissue, to serve as a proof of concept. As the inaugural investigation within this domain, it establishes a foundational framework for subsequent research endeavors.

## 5. Conclusions

We observed that indirect abnormalities in adipokine levels present substantial involvement in the inflammatory and metabolic pathways in the human organism and can affect the complexity, progression, and, therefore, treatment of cryptoglandular anal fistulas. Moreover, complex perianal cryptoglandular fistulas have a reduced level of the anti-inflammatory adipokine adiponectin and an increase in the levels of proinflammatory resistin and leptin. The above-described abnormal secretion of adipokines may affect the integrity of the ECM and the EMT in the fistula tract. E-cadherin, MMP2, and MMP9 expression levels were shifted in patients with more advanced and complex perianal fistulas. The presented results are less expressed and pronounced than in fistulizing Crohn’s disease; however, similarities in the pathogenesis are observed. 

## Figures and Tables

**Figure 1 ijms-25-01501-f001:**
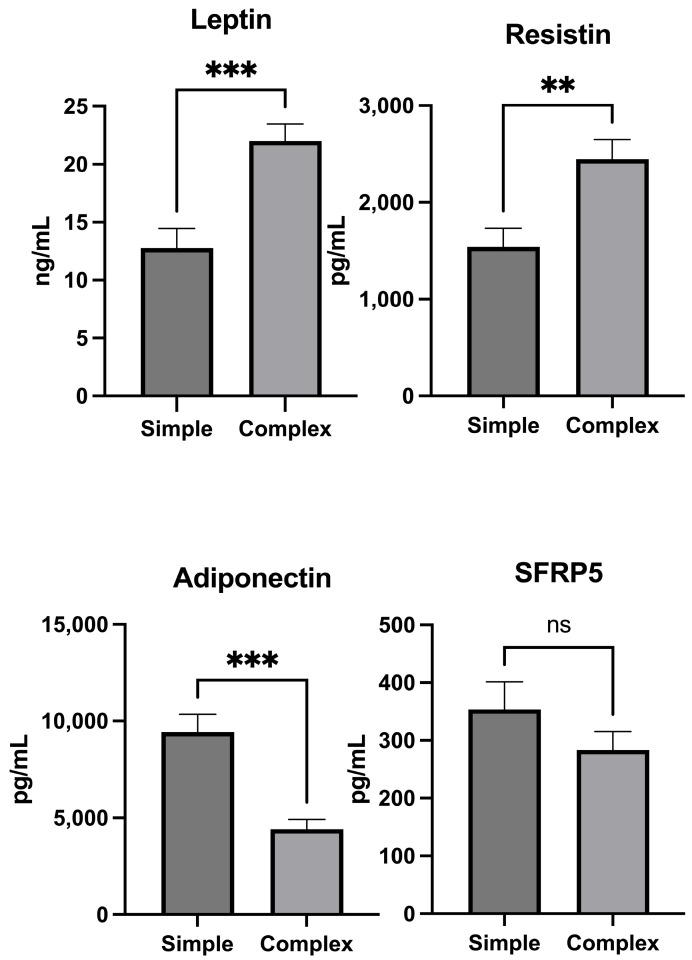
Adipokine levels in tissue samples from simple and complex cryptoglandular perianal fistulas. Data are expressed as mean ± SEM. ** indicates significant difference as *p* < 0.01, and *** as *p* < 0.001 and ns—indicates non-significant statistical differences.

**Figure 2 ijms-25-01501-f002:**
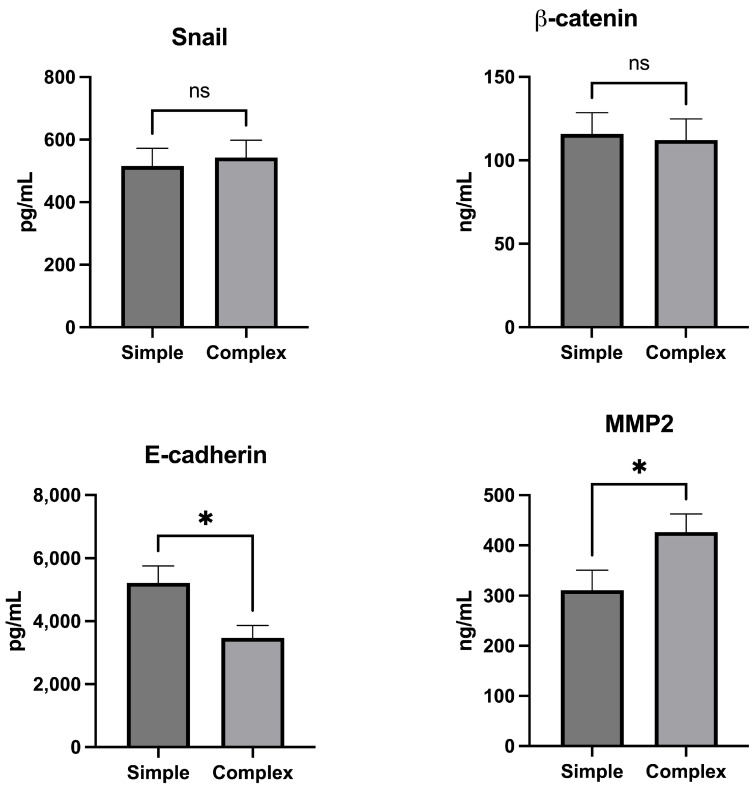
EMT and ECM integrity molecule levels in tissue samples from simple and complex cryptoglandular perianal fistulas. Data are expressed as mean ± SEM. * indicates significant difference as *p* < 0.05 and ns—indicates non-significant statistical differences.

**Figure 3 ijms-25-01501-f003:**
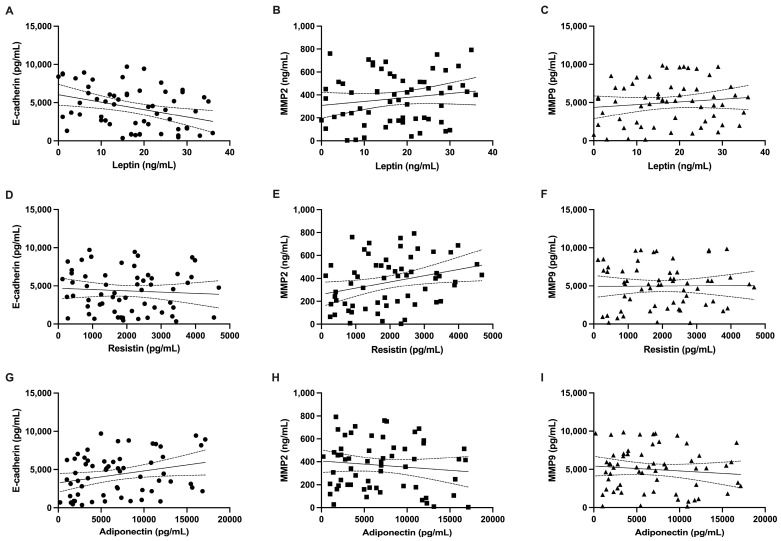
Correlation between tissue concentration of leptin (**A**) and E-cadherin (**B**) and between MMP2 (**C**) and MMP9. Correlation between resistin and (**D**) E-cadherin and between (**E**) MMP2 (**F**) and MMP9. Correlation between adiponectin and (**G**) E-cadherin and between (**H**) MMP2 (**I**) and MMP9. Data points (circles, squares, and triangles) represent each individual value.

**Table 1 ijms-25-01501-t001:** Characteristics of the study subjects.

Sex	Male	N = 31
	Female	N = 29
Age		43.04 ± 12.5
Fistula grade according to SPTF	Simple	N = 30
	Complex	N = 30
BMI	Simple	29.02 ± 2.85
	Complex	30.43 ± 3.71
Weight [kg]	Simple	74.14 ± 12.5
	Complex	78.93 ± 14.2

## Data Availability

Data available on request due to restrictions (e.g., privacy, legal or ethical reasons).

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
