# Peer review of "Role of Adipose Tissue Hormones in Pathogenesis of Cryptoglandular Anal Fistula"

_ijms, 2024, doi:10.3390/ijms25031501_

Round 1

Reviewer 1 Report

Comments and Suggestions for Authors

This is a study comparing the presence of several adipose hormone in simple or complex perianal fistula. The manuscript is well written and the topic is interesting but results are preliminary and based on a single technique ELISA of digested tissue. The authors should at least confirm their findings by other readouts: PCR, immunochemistry etc.

Moreover:

-How cytokines were chosen? Based on which studies?

-Figures 1, 2 3. Please change the size or orientation of the legend on axe x as simple and complex appear as a single word.

-Reference 2 did not demonstrate that fistuolotmy is the gold standard for fistula-in-ano and is only an audit on 600 patients operated in India. To the best of my knowledge, there is no clear consensus for the type of surgery.

-p.10 l.297 […]100lL[…] Please correct the typo

Comments on the Quality of English Language

The manuscript is well written. Minor editing required.

Author Response

Authors: We would like to appreciate the reviewer for his valuable insight and comments. We have addressed them and made proper corrections.

To the best of our knowledge this is the first study to analyze the concentration of adipokines in fistula tract tissue and their relation to the complexity of fistula. We have added the study limitation paragraph in our paper. We are aware that those results were based on single molecular technique, however we hope our paper will merit your expectations because of the novelty of this paper. Our goal is to further investigate this area of research and we have designed our methods to be more thorough.

Reviewer: -How cytokines were chosen? Based on which studies?

Authors: In the introduction section we have provided additional references, which suggested to us the cytokines that might be involved in this pathomechanism.

Reviewer:-Figures 1, 2 3. Please change the size or orientation of the legend on axe x as simple and complex appear as a single word.

Authors: We have changed the figures accordingly to this comment.

Reviewer:-Reference 2 did not demonstrate that fistuolotmy is the gold standard for fistula-in-ano and is only an audit on 600 patients operated in India. To the best of my knowledge, there is no clear consensus for the type of surgery.

Authors: Thank you for this comment. We have revised it and changed this paragraph in the introduction section.

Reviewer:-p.10 l.297 […]100lL[…] Please correct the typo

Authors: Corrected. 

Reviewer 2 Report

Comments and Suggestions for Authors

This is a good study in which the authors examined the role of adipocytokines in pathogenesis of cryptoglandular anal fistula. The results suggest that in adipose tissue resected from a perianal fistula, concentrations of proinflammatory cytokines were increased, while anti-inflammatory adiponectin  was decreased and that, together with changes in MMP2, MMP9 and E-cadherin concentrations, may be involved in cellular changes during fistula development.

The study is well conducted with no methodological flaws.

Before accepting the study's conclusions, a few things should be clarified:

1) The novelty that the study provides should be mentioned in the discussion.

2) The authors ignore the fact that not only adipocytes, but also infiltrating circulating monocytes and different amounts of macrophages in adipose tissue may be involved in the observed changes. Please explain.

3) It would be interesting if the authors provided data on the presence of obesity and plasma levels of adipokines in the tested patients. Does the increased pro-inflammatory environment associated with obesity affect the development of cryptoglandular perianal fistula?

4) Table 1 should be supplemented with more detailed characteristics of the group of patients, especially plasma levels of adipokines, which could be related to the concentrations in the perianal fistulas.

5) Lines 62 - 65: The sentence should be completed with a literary reference.

6) The authors should modify the statement in the first sentence in the Conclusion section. It is not possible to write: "We observed that abnormal accumulation of adipose tissue and..." when adipose tissue accumulation in the perianal fistula  or elsewhehre was not measured in the study.

7) All abbreviations should be explained the first time they appear in the text. For example:

89: Snail; 151: IBD patients; 153: DSS colitis;163: pERK ; 191 Wint; 208 :SFRP5; 255: JNK; etc.

Author Response

We would like to appreciate the valuable comments and insight provided by this reviewer.

Reviewer: 1) The novelty that the study provides should be mentioned in the discussion.

Authors: We have provided additional paragraph in the discussion section about the novelty of this study.

Reviewer: 2) The authors ignore the fact that not only adipocytes, but also infiltrating circulating monocytes and different amounts of macrophages in adipose tissue may be involved in the observed changes. Please explain.
3) It would be interesting if the authors provided data on the presence of obesity and plasma levels of adipokines in the tested patients. Does the increased pro-inflammatory environment associated with obesity affect the development of cryptoglandular perianal fistula?
4) Table 1 should be supplemented with more detailed characteristics of the group of patients, especially plasma levels of adipokines, which could be related to the concentrations in the perianal fistulas.

Authors: Thank you for this comment. Those are important aspect that should be properly addressed and investigated. We plan to further study the subject of adipose tissue and its' role in the perianal fistula and for sure we will analyze to impact of circulating monocytes. Currently we are starting the prospective study that will focus on the fistula healing, both plasma and tissue levels of adipokines, BMI, and the fat tissue distribution. However, we do hope that our study will merit your expectations in current version as it presents interesting proof-of-concept. We have added a paragraph regarding the limitations of our study.

Reviewer: 5) Lines 62 - 65: The sentence should be completed with a literary reference.
Authors: Corrected.

Reviewer: 6) The authors should modify the statement in the first sentence in the Conclusion section. It is not possible to write: "We observed that abnormal accumulation of adipose tissue and..." when adipose tissue accumulation in the perianal fistula  or elsewhehre was not measured in the study.
Authors: We have implemented correction accordingly to this comment.

Reviewer: 7) All abbreviations should be explained the first time they appear in the text. For example:
89: Snail; 151: IBD patients; 153: DSS colitis;163: pERK ; 191 Wint; 208 :SFRP5; 255: JNK; etc.
Authors: We have revised whole paper again and made adequate corrections.